# Toward Establishing an Ideal Adjuvant for Non-Inflammatory Immune Enhancement

**DOI:** 10.3390/cells11244006

**Published:** 2022-12-11

**Authors:** Tsukasa Seya, Megumi Tatematsu, Misako Matsumoto

**Affiliations:** 1Nebuta Research Institute for Life Sciences, Aomori University, Aomori 030-0943, Japan; 2Department of Vaccine Immunology, Hokkaido University Graduate School of Medicine, and Hokkaido University International Institute for Zoonosis Control, Sapporo 060-8638, Japan; 3Department of Medical Biology, Akita University Graduate School of Medicine 1-1-1 Hondo, Akita 010-8543, Japan

**Keywords:** Toll-like receptor 3 (TLR3), dendritic cells, cross-antigen presentation, TICAM-1 (TRIF) pathway, Th1 polarization, vaccine

## Abstract

The vertebrate immune system functions to eliminate invading foreign nucleic acids and foreign proteins from infectious diseases and malignant tumors. Because pathogens and cancer cells have unique amino acid sequences and motifs (e.g., microbe-associated molecular patterns, MAMPs) that are recognized as “non-self” to the host, immune enhancement is one strategy to eliminate invading cells. MAMPs contain nucleic acids specific or characteristic of the microbe and are potential candidates for immunostimulants or adjuvants. Adjuvants are included in many vaccines and are a way to boost immunity by deliberately administering them along with antigens. Although adjuvants are an important component of vaccines, it is difficult to evaluate their efficacy ex vivo and in vivo on their own (without antigens). In addition, inflammation induced by currently candidate adjuvants may cause adverse events, which is a hurdle to their approval as drugs. In addition, the lack of guidelines for evaluating the safety and efficacy of adjuvants in drug discovery research also makes regulatory approval difficult. Viral double-stranded (ds) RNA mimics have been reported as potent adjuvants, but the safety barrier remains unresolved. Here we present ARNAX, a noninflammatory nucleic acid adjuvant that selectively targets Toll-like receptor 3 (TLR3) in antigen-presenting dendritic cells (APCs) to safely induce antigen cross-presentation and subsequently induce an acquired immune response independent of inflammation. This review discusses the challenges faced in the clinical development of novel adjuvants.

## 1. Introduction

The genomic DNA of vertebrates (including humans) mutates over time and thereby evolves. On the one hand, DNA mutates endogenously at a certain frequency during the replication process, regardless of survival advantage or disadvantage. Mutations are often neutral. In general, genetic drift is subject to environmental selection, and favorable traits are fixed. On the other hand, genomic DNA can also mutate in response to exogenous factors. Exogenous genetic mutations include cases in which infection is mediated by foreign agents such as retroviruses or transposons. The invasion of microorganisms and foreign genes threatens the identity of an individual’s DNA, and the host immune system defends against this crisis by eliminating non-identical nucleic acids and proteins by inducing cell death and inflammation.

The immune system is the general term for the mechanisms that identify and eliminate the presence of the ‘non-self’. Mutations can lead to evolution if they occur in germ cells and to cancer if they occur in somatic cells. Innate immunity prevents microbial invasion and specifically recognizes nucleic acids and membrane components, while acquired immunity specializes in identifying mutations in proteins and peptides. Both have surveillance mechanisms and appear to be sophisticated self-preservation mechanisms. Immune enhancement is one strategy for controlling infectious diseases and cancer [1,2]. Vaccination is a method of immune enhancement that targets certain ‘non-self’ organisms, usually by means of antigens combined with adjuvants in order to eliminate them [1,2]. Substances with an immune-enhancing role are called adjuvants. Adjuvants are a very important component of vaccines, but their own immunopotentiating effect (without antigen) is difficult to assess ex vivo and in vivo. The host response to an adjuvant usually reflects its inflammatory features rather than its immunopotentiating effect. Therefore, there are few safe and effective general-purpose vaccine adjuvants and none have been established in drug discovery research.

ARNAX is a synthetic double-stranded RNA hybrid molecule capped with DNA [3]. The molecule is chemically engineered to be relatively resistant to nucleases; ARNAX was endocytosed into Toll-like receptor 3 (TLR3)-positive cells simply by adding to the cells (without transfection reagents), and the dsRNA portion in ARNAX-activated TLR3, which exclusively induced signal transduction of the Toll/IL-1R homology (TIR) domain-containing molecule 1 (TICAM-1) pathway [3]. Efficient TLR3 activation by ARNAX required a 120–140 bp double-stranded RNA portion [3,4]. No activation of cytoplasmic RNA sensors was observed in cells sprinkled with ARNAX [4]. In dendritic cells that responded to ARNAX, levels of the co-stimulators, interferon (IFN)-message and IL-12p40 were increased (Table 1) [5,6,7]. ARNAX served as an effective immunopotentiator in animal models with implant tumors or viral infections without toxic side effects upon i.p. or s.c. injection [3,4,8]. The properties of ARNAX appear to be more reasonable than the early reported double-stranded RNA adjuvants polyI:C and polyI:C(12U) in terms of cytokine toxicity (Table 2). Thus, ARNAX would be a candidate for a safe Th1-skewing adjuvant that overcomes regulatory approval. This paper reviews the properties of ARNAX compared to current Th1 adjuvants and discusses the safety and efficacy of adjuvants in vaccine development.

## 2. Innate and Acquired Immunity

The immune system is classified into innate and acquired immunity. Innate immunity recognizes components of microorganisms (microbe-associated molecular patterns, MAMPs, pathogen-associated molecular patterns, PAMPs) and regulates cells through receptor signaling; MAMPs/PAMPs are characteristic motifs present in microbes but not in humans and are identified by host pattern recognition receptors (PRRs) [9]. Innate immune PRRs, represented by Toll-like receptors (TLRs), are conserved in almost all multicellular organisms [10,11]. PRRs recognize “non-self” nucleic acids and signal the exclusion of foreign organisms even in plants [12], particularly protecting the host from infectious pathogens. Thus, MAMPs are potential candidates for vaccine adjuvants, some of which are being investigated in clinical trials. However, a series of innate immune signals induced by MAMP administration, accompanied by severe inflammatory toxicity and disease signs and symptoms, forced the discontinuation of clinical trials. Insofar as vaccine side effects are representative of the innate immune response, MAMP administration would mimic the symptoms of infectious disease. In other words, the genomic and nucleic acid identity of multicellular organisms is maintained by acute inflammation caused by the host’s innate immunity. This highlights the fact that the use of MAMPs as vaccine adjuvants must be devised in some way to ensure their safety prior to clinical use.

In humans, the innate immune response of the host varies widely among individuals. In addition to genetic factors such as single nucleotide polymorphisms, epigenetics plays a major role in orchestrating the biological response to pathogens, including changes in (or activation of) cells responsible for immunity and cytokine responses [13,14]. In other words, infectious diseases are systematized as syndromes caused by host responses triggered by innate and acquired immunity in addition to pathogen virulence [15]. Thus, the challenge is to design adjuvants that evoke the immune system harmlessly rather than inducing the pathogenicity of infectious diseases.

Innate immunity can be triggered by all cells in the body after exposure to microorganisms and is not restricted to immune cells. Typical type I transmembrane receptors that sense infection by foreign pathogens are TLRs, while cytoplasmic receptors that sense intracellular infectious pathogens are RIG-I-like receptors (RLRs) and Nod-like receptors (NLRs) [11]. These are collectively referred to as PRRs [9] and are distributed in cells throughout the body, resulting in a systemic response to infection; PRRs simultaneously function as adjuvant receptors. Thus, microbial adjuvants are presumed to be involved in the pathogenesis and development of infections. For this reason, many attenuated vaccines that mimicked infectious diseases must have reached a high threshold of safety.

Activation of acquired immunity is achieved by mechanisms involving antibody production or activation of T lymphocytes [1,2,15]. The current consensus on the origin of the immune system is that innate and acquired immunity are separate systems but hierarchically linked in vertebrates [16]. Acquired immunity is essentially a mechanism by which protozoa selectively produce various surface proteins by recombination or transformation of genes, as indicated by the presence of rearrangement-activated genes (Rag) and transposase activity in protozoan prototypes of surface protein change mechanisms [17]. Protozoa evade attack by host immune cells by exchanging surface proteins. Human-acquired immunity is a derivative of a gene rearrangement mechanism incorporated into lymphocytes, the activation of which has multiple triggers independent of infection. Acquired immunity is defined as an induced response triggered by interferons and accompanied by inflammation [18]. In humans, acquired immunity occurs simultaneously with or secondary to innate immunity [9,11,16,19]. This sequence of immune activation is conserved from lamprey to humans.

Activation of acquired immunity is characterized by the proliferation of specific lymphocyte clones triggered by signals from cell adhesion clusters known as antigen-presenting complexes [20]. Dendritic cells select specific T lymphocytes and subsets to be responsible for antigen presentation [5]. That is, the activation of acquired immunity can be monitored by the proliferation of T cells and subsets with receptors that match the specific antigens presented by the major histocompatibility complex (MHC) of dendritic cells [5,21]. Activation of T lymphocytes reflects the binding of dendritic cells to T cell molecular clusters. Molecules and cytokines that join clusters on dendritic cells tend to be upregulated in response to adjuvants, leading to amplified T-cell activation [9,11,20].

B lymphocytes are responsible for antibody production; B lymphocytes have antigen receptors that recognize antigens and activate specific B-cell clones [22]. Through dendritic cell-dependent or -independent pathways, B cells are potently activated (antibody production) [23]. Activation-induced cytidine deaminase (AID) governs B-lymphocyte class-switching (switch recombination), and humoral factors and CD4+ helper T cells are involved in B-lymphocyte activation [23,24]. Adjuvants are thought to be important in enhancing the production of subclass-specific immunoglobulins by promoting class switch recombination [22,25].

Natural killer (NK) cells belong to the innate lymphocytes and are activated in response to adjuvants [3,26,27]; NK cell activation involves dendritic cell-dependent and -independent pathways, and the Rae-1 molecule of stimulatory cells appears to be involved in NK receptor activation [26,27]. Adjuvants may be associated with the activation of other innate lymphocytes, NKT cells, and γδT cells, which are ongoing topics [28,29,30]. Vaccines could be designed to selectively reproduce this sequence of events, and adjuvants could help to achieve these cascades.

## 3. Antibodies and T Cells in Viral Infections

Dendritic cells bind T cells displaying the MHC complex (CD4/8, MHC (peptide)–T cell receptor, TCR)) and other co-stimulatory molecules [31,32]. The dissociation constant (Kd) of the MHC (peptide)–TCR complex is weak, about 10^−6^ (depending on the sequence), and requires other molecules for strong adhesion [15]. Furthermore, a large number of TCRs must be assembled to allow sufficient proliferative T-cell signaling [15]. CD8+ T cells activate proliferation pathways, such as Src, protein kinase C (PKC) and PI3-kinase, through dendritic cell–T-cell adhesion. Neutralizing antibodies recognize antigen structures with four planes in the variable region and thus have high affinity, with the Kd of an antibody–antigen complex usually greater than 10^−9^ [15]. In chimeric antigen receptor T (CAR-T) cells, antibody F (ab) have been shown to bind well to CD19 and other molecules on target cells by external cross-linking [33]. Antibodies are often effective in recognizing not only antigens on cells but also free antigens in the liquid phase [34,35]. The multifaced recognition shows the advantage of neutralizing antibodies. The advantage of cytotoxic T lymphocytes (CTLs) lies in recognition of intracellularly processed short-chain peptides. Short-chain peptides were conceived as candidates for universal vaccines against cancer and infectious diseases. Mutant antigens, testis-specific antigens, and pan-cancer antigens, such as WT1, have been selected as cancer vaccines. However, exogenous administration of short-chain antigen peptides is usually unsuccessful because neither phagocytosis nor antigen cross-presentation occurs [36]. Furthermore, the diversity of the human MHC creates additional challenges. Because humans are hybrids, it can be difficult to extrapolate results from inbred strains of mice to human clinical trials [36].

In general, the efficacy of vaccine-induced antiviral immunity depends on the type of virus, even when the vaccine-associated enhanced disease is excluded [37]. For certain viruses, such as respiratory syncytial virus (RSV) and SARS-CoV, vaccine efficacy cannot be predicted from the antibody titer of vaccines [38]. In such cases, evaluation of “antiviral effectors” other than antibody titers is necessary for prevention [39,40], and fever and inflammatory signs are not necessarily reliable markers for assessing vaccine efficacy.

What is needed for the SARS vaccine is not an increase in neutralizing antibody titers but activation of dendritic cells by Th1 adjuvants [38,39]. This includes combined effects consisting of antigen-specific CTL induction, CD4+ T cell activation, and antibody production. In addition, other inducers for dendritic cell activation may exist. Where vaccines have proven to be effective, for example, against measles, mumps, and rubella viruses, a distinction must be made because neutralizing antibodies are effective against these viral infections [15]. In addition, antibody titers alone may not be sufficient to determine vaccine efficacy [40,41,42]. For this reason, viral vaccines require the quantification of CTLs as well as antibodies.

## 4. Cross-Antigen Presentation of Extrinsic Antigens

Infection presents the host immune system with antigens unique to the microorganism, the antigens being released from the infected cells. Antigens are taken up by dendritic cells from foreign sources (Figure 1), except in rare cases where the pathogen directly infects dendritic cells (e.g., measles infection) [43]. Foreign antigens are usually presented on the surface of dendritic cells by class II presentation [5,15]; CD4+ T lymphocytes recognize the MHC class II-peptide complex in dendritic cells and are activated (Table 1). Notably, foreign antigens from infected cells are taken up by professional antigen-presenting dendritic cells (mouse CD8α+ DC, CD103+ DC, human CD141+ DC) and presented to CD8+ T lymphocytes by MHC class I presentation. Another mechanism known as the cross-antigen presentation can occur in response to uptake antigens [44,45], and notably, the action of PAMPs (=adjuvants) experimentally promotes cross-antigen presentation [46,47] (Table 1). Phenotypically, after infection or vaccine administration, lymphocytes proliferate (i.e., lymph node swelling), and specific antibodies and CD8+ T lymphocytes increase in response to the antigen/adjuvant complex, but not either alone [48]. Microorganisms are eliminated only as a result of the immune-boosting effects of the antigen/adjuvant combination. Lymphocyte proliferation may interfere with the proliferation and metabolism of other cell types but is not in itself a safety concern. Thus, adverse vaccine events are largely defined by the innate immune response induced by the adjuvant.

This figure shows a possible route of cross-antigen presentation hypothesized to be caused by dendritic cells that have phagocytosed foreign substances (antigens). Foreign proteins are degraded by proteases, such as cathepsins, and are trimmed by insulin-regulated aminopeptidase (IRAP) and other enzymes in (phagolysosomes or) endosomes. MHC I molecules are recruited to this endosome either from the plasma membrane or from the ER. The degraded/trimmed peptide is mounted on MHC I and further trimmed by endoplasmic reticulum aminopeptidase (ERAP). MHC I/peptide complexes are transported on the cell membrane. This pathway is called cross-antigen presentation because it differs from normal MHC I presentation. Normally, endogenous peptides that have been degraded by proteasomes in dendritic cells drop into the endoplasmic reticulum (ER) in a transporter-associated antigen-processing (TAP)-dependent manner and localized MHC I loads the peptides and move to the cell membrane. Peptides can be generated using a TAP-independent mechanism. Adjuvants facilitate the mechanism of cross-antigen presentation of phagocytosed foreign antigens. The TLR3-TICAM-1 pathway experimentally promotes the presentation of foreign antigens, but the mechanism of cross-antigen presentation upon in vivo stimulation by TLR3 agonist alone remains to be addressed (dashed lines).

## 5. Recognition of Subthreshold Antigens by Dendritic Cells with Adjuvant

With an adjuvant alone, dendritic cells do not mount an immune response [48,49]. However, repeated stimulation of PRR accelerates the immune response of dendritic cells even in the absence of antigens and potentiates the response to antigens encountered in the future [50,51]. This is explained by changes in dendritic cell epigenetics and metabolism [51] and a concept known as trained immunity [52]. TLRs are important triggers for initiating trained immunity [48,50,51]. Repeated stimulation of TLRs in dendritic cells enables these cells to sense barely detectable amounts of antigen. Moreover, the signal response of TLR becomes amplified compared with the first stimulation. However, the establishment of trained immunity is accompanied by a number of risks, including inflammatory side effects, exhaustion and the exacerbation of autoimmune diseases.

Innate immune activation can potentially induce acquired immunity. It has been shown that repeated innate stimulation provides a method for preventing latent cancer and subclinical infections, as immune surveillance facilitates their clearance. If a TLR-specific adjuvant successfully activates dendritic cells without involving other cells of the body, this enables the dendritic cells to respond to lesser antigen signals without side effects [53,54]. Most TLRs and PRRs are, however, broadly distributed across human tissues and organs, with the exception of TLR3, TLR7, and TLR9. TLR7 and TLR9 are exclusively expressed in plasmacytoid dendritic cells (pDCs), which mainly serve as cytokine inducers with a lesser ability for cross-antigen presentation [55,56]. It is TLR3 that confers potent cross-antigen presentation in dendritic cells [53,54]. Since human antigen-presenting dendritic cells (represented by CD141+ cells) predominantly express TLR3 [57], TLR3-specific ligands may achieve this goal of amplified antigen-sensing without adverse toxicity [53,54,58]. Activation of PRRs by such non-inflammatory agonists induces immune cell recruitment, signaling, and the secretion of appropriate effector molecules, inducing favorable immune responses to mitigate invading pathogens or cancers [58,59]. The administration of a safe adjuvant may be a prerequisite to appropriately sensitize the initial response of acquired immunity.

## 6. TICAM-1 Pathway in Dendritic Cells

Dendritic cells can be discriminated into multiple subsets based on their surface markers. Direct stimulation of TLRs of specific dendritic cell subsets is an efficient method of cross-antigen presentation [60,61]. However, TLRs that are predominantly expressed by dendritic cells must be selected. Both mouse CD8α+/CD103+ and human CD141+ dendritic cells (professional antigen-presenting cells, APCs) highly express TLR3 [57,62,63], which binds to TICAM-1 but not to the MyD88 adapter [11,64], suggesting that the TLR3- TICAM-1 pathway can achieve this goal [65]. TLR3 is highly expressed in APCs and low in other cell types except epithelial cells [66,67]. If so, targeting TLR3 in dendritic cells with specific adjuvants would minimize cytokine toxicity of the systemic response, minimize effects on other cells [68,69], and raise antigen-specific T lymphocytes without signs of inflammation (Figure 2). Notably, all human TLRs except TLR3 are linked to MyD88 and induce inflammatory responses [70], and a TLR3-specific adjuvant (herein referred to as ARNAX) is currently under development based on a nonclinical proof of concept (POC) [65].

This figure depicts the outcome of ARNAX-dependent TLR3 activation in APCs (e.g., human CD141+ DCs). TLR3-specific activation by ARNAX does not increase cytokine/IFN levels in blood circulation. Cytokinemia is usually induced secondary to TLR stimulation of other infected cells in the body. Dendritic cells are activated either by adjuvant with DC-targeting or inflammation. Red arrays indicate the DC-targeting pathway, and black arrays the inflammatory pathway induced by viruses, targeting various cells in the body. In response to ARNAX, three representative immune cell activations are observed without inflammatory signs: Th1 polarization, CD8+ T cell activation, and Ab production. DC-targeting results in DC-mediated activation of NK cells, although they belong to innate lymphocytes. How ARNAX regulates other lymphocyte moieties, including γδT or NKT cells, remains elusive. The DC-targeting pathway is superior to the inflammatory pathway in that it elicits cellular immunity with fewer adverse events. 

Another adjuvant called monophosphoryl lipid A (MPLA), which activates the TLR4-TICAM-1 pathway, has been approved as a low-toxicity adjuvant [71] and is currently applied in several vaccines (such as GSK AS02 and AS04) [72]. TLR4 ligands activate two adaptors (MyD88 and TICAM-1) that are differentially biased against the drug (Table 2); MPLA favors TICAM-1 over MyD88 [65,70]. TLR4 is expressed in mouse APCs but not in human APCs [57]. TLR3 and TLR 4 bind with different ligands but have in common the activation of TICAM-1. Both TLR3 and TLR4 agonists select for the TICAM-1 adaptor, making them suitable as adjuvants to minimize inflammation induction and promote antigen presentation (Figure 1 and Figure 2). TLR4 is not expressed in human APCs but is expressed in other dendritic cell subsets and macrophages [73]. TLR4, unlike the case of TLR3, requires TICAM-2 (TRAM) for activation of TICAM-1 [11,47,70].

No other examples of TLRs adopting TICAM-1 as an adaptor have been reported so far; STING may be regulated by TICAM-1 in a limited variety of cells [74,75]. TLR7 and TLR9 are expressed specifically in pDC and adopt MyD88 as an adaptor [55,76]. pDC is a potent inducer of type I interferon (IFN) and cytokines, inducing systemic inflammation, but it has a low capacity for cross-antigen presentation [56,77]. However, there are opposing views as to pDC antigen presentation capacity in several papers [77,78]. Cross-antigen presentation may occur under the activation of MyD88 in other dendritic cell subsets; TLRs that use MyD88 as an adaptor is not limited to APC expression [11], and safety issues, along with the side effect of systemic inflammation, are barriers and have not been approved under the Pharmaceutical Affairs Law, except TLR4. Thus, the TICAM-1 pathway would provide the conditions for a safe dendritic cell adjuvant.

## 7. Cross-Presentation Induced by the TICAM-1 Pathway in Dendritic Cells

TLR3 is highly expressed on antigen-presenting dendritic cells (CD11+ DCs) [57] and initiates the mechanism of foreign antigen presentation (called cross-antigen presentation) (Figure 1); TLR4 is also expressed on other CD1c positive dendritic cells and macrophages [75], in a process called dendritic cell maturation via the TICAM-1 pathway, conferring direct antigen-presenting capacity to dendritic cells.

TICAM-1 signaling activates downstream transcription factors NF-kB, IRF-3, and AP-1 to promote their nuclear translocation in immune cells [58,64,79]. In other cells, including tumor cells, TICAM-1 signaling activates the RIP-1/3 pathway and also promotes IRF-3 transcription and type I IFN induction [80,81,82]. TICAM-1 appears to induce less NF-κB activation than MyD88 [64,79]. Transcription factors of the AP-1 family are surmised to be strongly involved in cross-antigen presentation [83]. SEC61 is also reported to be downstream of IRF-3 and involved in cross-antigen presentation [84]. Tlr3˗/˗, Ticam1˗/˗, Irf3˗/˗, and Ifnar˗/˗ mice lose the ability to induce proliferation of CD8+ T lymphocytes in antigen presentation [4]. Positive feedback amplification of type I IFN by lymphocyte IFN-γ production and dendritic cell IFNAR is thought to promote cross-antigen presentation [85].

Compared to other RNA adjuvants, ARNAX is an adjuvant that specifically activates only TLR3 during cross-antigen presentation (Table 2). Any combination of antigens can constitute a vaccine; even if ARNAX were introduced into humans, trained immunity could be safely activated. A nontoxic trained immune response targeting CD141+ dendritic cells would trigger cross-antigen presentation in response to reduced antigen levels.

## 8. Adjuvants for Cancer Vaccines

It was first reported in 1990 that cancer cells express specific antigens [86]. Hundreds of tumor-associated antigens (TAAs) have since been identified [87,88]. This raised the question as to whether acquired immunity recognizes cancer-specific antigens as “nonself” and eliminates them during so-called, antitumor immunity. For more than 30 years, peptide vaccine therapy using cancer antigens has been investigated; however, clinical trials have been unsuccessful to date [89]. In 2012, it was demonstrated that cancer immunity plays a role in cancer treatment as a surveillance mechanism by inhibiting immune checkpoints, which is different from the strategy involved in vaccines [90]. Acquired immunity can successfully eliminate not only infectious agents but also cancerous cells.

One possible reason why cancer vaccines have been unsuccessful to date may be that peptide vaccine were administered without immune enhancers (adjuvants), unlike vaccines for infectious diseases. It is worth noting that 25% of cancers originate from infection, and an understanding of infectious disease and the use of adjuvants in the context of trained immunity may be crucial in the development of anticancer therapies [91].

Previously, peptide vaccine therapy for cancer has predominantly involved the subcutaneous administration of cancer antigens, e.g., 8–10 amino acids (CD8 epitope) or ~13 amino acids (CD4 epitope) [89]. A method of directly administering the antigens to patients and observing tumor regression was adopted as an efficacy marker. Two main problems were encountered: (1) dendritic cells do not present antigens simply by exposure to exogenously administered peptides [89], and (2) the adjuvant used was inadequate [92,93]. In many cases, the peptide was mixed with oil (montanide or squaran) to enhance immunity locally without stimulating cell proliferation. Initially, there were no approved adjuvants other than alum, which was contraindicated as a result of its potential to drive Th2 polarization [94,95,96], which promotes tumor growth [97]. Alum may activate NALP3 inflammasome to deteriorate tumor microenvironment (TME) [98,99,100]. In addition, the results of non-clinical tests using animal models were not directly applicable to human clinical trials because the antigens differed from those in humans. Furthermore, the administration of an antigen alone does not mimic pathogen infection, and, in the same way, a single antigen only cannot trigger anticancer immunity by T cells [53,54].

Many infections are cured following an immune response. If cancer immunity is to mimic the effects of infectious disease vaccines, the antigens (either mutant antigens or testis-specific antigens) must trigger immunity, particularly with the mutations or altered expression levels of the antigen, and epitopes must be targeted for MHC presentation despite individual differences.

Solid tumors present a tissue microenvironment that may be refractory to treatment [98,99], somewhat resembling cases of tuberculosis or DNA virus infections. Chemotherapy has been established as a superior treatment to immunotherapy in infectious diseases and is the preferred therapy being actively considered in anticancer therapy. It is difficult to define the functional profile of TME in each tumor that comprises variable rates of stromal cells and cancer cells [92,98,99]. The stroma contains environmental factors such as regulatory T cells (Treg), myeloid-derived suppressor cells, neutrophils, cancer-associated fibroblasts, and tolerogenic dendritic cells that control immune activation in a complex manner [92,98]. These immune-regulatory factors, as well as a cytokine environment induced by dead cells, may affect the properties of TME in individual tumors [98,99]. Inflammation in cancer is generally controlled by environmental factors, including TME [92,98,99]. Acquired immunity is affected by both activation and exhaustion and is also affected by the time axis for the construction of the TME.

## 9. Adjuvant in Prophylactic Vaccine against Infectious Diseases

Vaccines are one method of deliberately activating immunity to protect against infection. A typical vaccine consists of an antigen and an adjuvant, and immune enhancement usually results from the addition of an adjuvant. However, live vaccines, inactivated whole-particle vaccines, and mRNA vaccines are examples where endogenous adjuvants have ensured immune enhancement. In general, attenuated and inactivated viral and bacterial vaccines contain live adjuvants, the amount of which often cannot be adjusted, resulting in lot-to-lot variation in vaccines. This inevitably results in adverse reactions related to infection. Such side effects often result in candidate vaccines failing toxicity tests, and, therefore, not gaining approval. Many of the licensed routine and optional vaccines use alum as an adjuvant, purportedly from a conventional point of view. The advantages of alum in prophylactic vaccines are that it barely promotes serious fever and inflammation, and does not affect the impurity of the antigen preparation (including the PAMP components). However, despite its long-term use, the molecular response to alum has not been fully defined [95,96], making it impossible to clearly define and standardize adjuvant function [101,102]. Although alum has been used in many vaccinations without serious troubles, there are several counter-reports on vaccine applications. For example, alum in the human papillomavirus vaccine preventing uterine cervical cancer might participate in a chronic fatigue-like syndrome according to several reports [103,104].

Immune responses to vaccines vary from individual to individual and between infectious and mutant strains. Vaccines can also induce adverse events, such as an over-amplified immune response, which can cause damage to the host [105,106]. In general, vaccines require immune-boosting by the inclusion of safe adjuvants. It is essential to establish non-toxic and versatile adjuvants that can be used against a wide range of antigens in all patients. Next-generation vaccines that are superior to mRNA vaccines need to be supported by the development of adjuvants that meet safety standards.

## 10. Perspectives

Adjuvants include a wide range of compounds recognized by PRRs. TLR signals (with the exception of TLR3) converge onto MyD88 [11,70] and are classified as inflammatory adjuvants. Only TLR3 can induce selective activation of antigen-presenting dendritic cells through TICAM-1 alone [64,79] and is thereby considered a safe dendritic cell adjuvant. Thus, adjuvants can be categorized into inflammatory adjuvants and dendritic cell adjuvants by clear definitions, and the current categorization based on identified and unidentified receptors is merely a historical convenience [107]. Adjuvants should be defined based on their comprehensive analysis [108,109], molecular interaction and activation of the immune system, especially involving innate immune receptors (PRRs), and it is necessary to evaluate their functional identification, efficacy, and safety separately. It is molecularly understood that adjuvants command dendritic cells to activate the main response, acquired immunity, while the accompanied induction of excessive inflammation and cytokine storms is a separate side response.

PLGA-Riboxxim, a recently published dsRNA adjuvant, were created under a conceptualization similar to ARNAX. PLGA-Riboxxim carries a TLR3/RIG-I ligand, and targets DCs to potentially activate therapeutic immunity [110]. If the additional RIG-I activation works better for diseased patients, this type of immune-enhancing strategy may be advantageous to protect patients, particularly vulnerable older populations, from infection or cancer.

Our proposal is to enhance trained immunity with dendritic cell adjuvants such as the TLR3 agonist ARNAX to lower the antigen threshold for detecting latent infection and cancer and facilitate the creation of more sensitive and effective preventive vaccines. Guidelines and potential biomarkers for the development of next-generation adjuvants have not yet been provided [107]. Given that the host response to adjuvants reflects the natural response to foreign nucleic acids/proteins, pharmaceutical law needs to reasonably relax restrictions on adjuvant approval. Based on knowledge of the structural–functional relationship of MAMPs/PAMPs and modification of dendritic cell-targeted reagents, it is possible to create a new class of adjuvants based on the active artificial adjuvants with the removal of the inflammatory component—a forthcoming consideration for the development of ideal vaccines.

## Figures and Tables

**Figure 1 cells-11-04006-f001:**
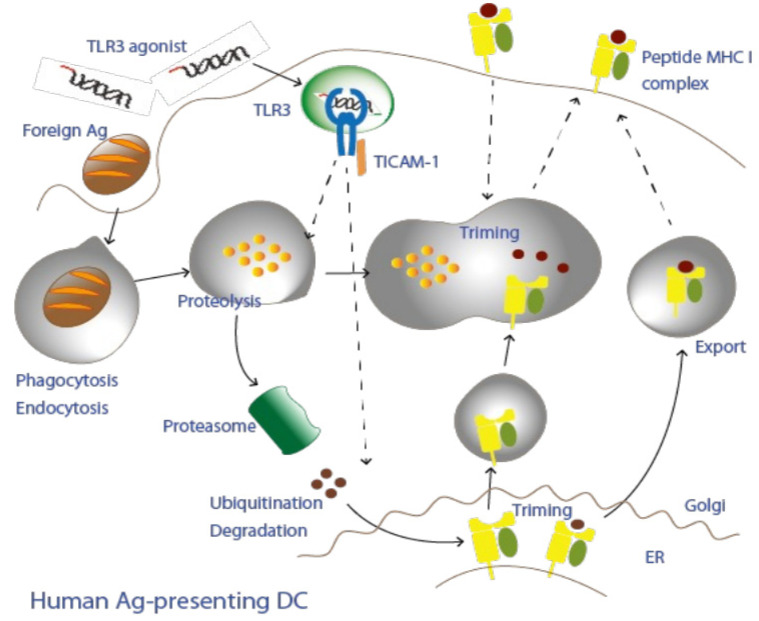
A putative model for antigen presentation in CD141+ dendritic cells.

**Figure 2 cells-11-04006-f002:**
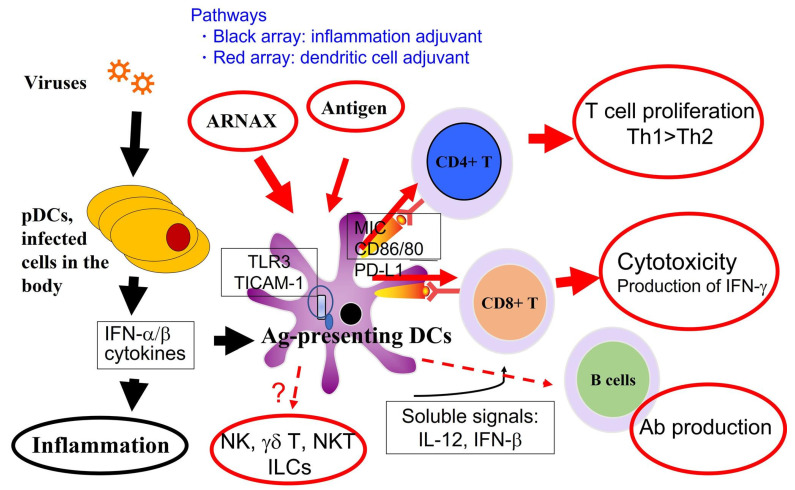
Acquired immune response induced by TLR3/TICAM-1-mediated dendritic cell activation.

**Table 1 cells-11-04006-t001:** Properties of antigen presentation.

Presentation	Class I	Class II	Cross-Present
Origin of the peptide	Intracellular	Extracellular	Vacuolar/Cytosolic
Localization	Proteasome	Endosome	Phagosome
Expressing cells	All host cells	Immune cells	DCs
Invariant chain	No	Yes	No
T cell binding Response to ARNAX	CD8 in CTL Unknown	CD4 in helper T Unknown	CD8 in CTL Up-regulated

**Table 2 cells-11-04006-t002:** Properties of the TICAM-1 adjuvants.

Ligand	Receptor	Adapter	Target Cells		Outcome		
			Mouse DC	Human DC	Skewing	Cross-Present	Cytokine Storm
ARNAX	TLR3	TICAM-1	CD8α DC CD103 DC	CD141 DC	Th1	+++	-
Ampligen	TLR3/MDA5	TICAM-1 > MAVS	Not restricted		Th1	++(?) *	+(?) *
PolyI:C	MDA5/TLR3	TICAM-1 < MAVS	Not restricted		Th1	+++	+++
MPLA	TLR4	TICAM-1 > MyD88	CD8a DC CD103 DC	CD1c DC	Th1	++	++

* (?) No concrete data have been reported or available.

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
