# Peer review of "Toward Establishing an Ideal Adjuvant for Non-Inflammatory Immune Enhancement"

_cells, 2022, doi:10.3390/cells11244006_

Round 1

Reviewer 1 Report

This is an important contribution looking at the role of adjuvants in vaccine development.. 

It will be helpful if you add a paragraph/section on adjuvants for infectious diseases.

Author Response

We appreciate the comments raised by Reviewer 1.

  • It will be helpful if you add a paragraph/section on adjuvants for infectious diseases.

We discussed adjuvants in the context of infectious diseases in the section “Adjuvant in prophylactic vaccine against infectious diseases” from line 371. There are a number of infectious diseases in human, and vaccines have been devised for each disease individually. However, this is not a review for expounding at length on the subject of vaccines in each disease. We mention general problems in vaccine adjuvant currently adopted: Alum’s problem and general side effects. Of course, we have no intention to neglect the fact that many case-by-case problems and adverse events happen in administration of adjuvant. Although these are important issues in vaccination, the issues appear somewhat beyond the purpose of this compact review.

Reviewer 2 Report

This review discusses the adjuvant effects and related issues mainly on a TLR3 agonist, ARNAX. Overall this review seems interesting for readers in immunology and vaccine fields and deals with recent progress in this field.

Major comment

1. It may be better to describe more on characteristics of TLR3 ligands, especially of ARNAX. For example, what kinds of RNAs are recognized by TLR3? In addition to ARNAX, there should be several TLR3 agonists, such as poly(I:C) or poly(I:C)12U (Ampligen). How are they and their functions similar or different? Especially, what are the features, if any, of ARNAX, compared with the other TLR3 ligands? I suggest authors to describe more on these points.

Author Response

We appreciate the comments raised by Reviewer 2.

  • It may be better to describe more on characteristics of TLR3 ligands, especially of ARNAX.

We agree to this comment. A short overview on ARNAX was added in the Introduction (line 57-72). Several comparative reviews on polyI:C, ampligen and ARNAX have been reported so far (Ref. 50, 106, 107), and we cited these reviews in our present review.

Actual comparative experiments with ampligen and ARNAX in parallel are difficult to execute. No one can do so without contract. The name PolyI:C does not define the structure or length of polyI and polyC, but different batches are regarded as the same without distinction. This remains to be a barrier of the comparative studies, also.

Reviewer 3 Report

Seya et al. review the role of adjuvants and the underlying mechanisms with a focus on cross-antigen presentation by dendritic cells. Additionally, the review discusses some of the issues facing the clinical development of novel adjuvants.

This is an interesting review highlighting the potential benefit of using a dendritic cell adjuvant, such as a TLR3 agonist, ARNAX, that is a non-inflammatory adjuvant candidate. The authors have previously carried out several pre-clinical studies on investigating the adjuvant effects of ARNAX in cancer, influenza and COVID-19 vaccines and have published several articles on ARNAX adjuvant. However, some of those key details are missing in this review and the importance/significance of ARNAX needs to be briefly defined at the onset of the paper (in the summary and introduction sections) to make a final recommendation. Additionally, the following points need to be addressed to improve the quality of the paper.

1.     Is the title reflecting the main concept of the review paper? The current title seems to reflect just one aspect of the entire paper. Suggest rewriting the title to better cover the contents in this review. 

2.     The title mentions “ARNAX vaccine”, which is misleading. ARNAX is a vaccine adjuvant, not a vaccine. Please rewrite for clarity.

3.     There should be a brief mention about ARNAX adjuvant in the summary section for coherence with the main review point in this manuscript.

4.     Line 53-54: Does not mention anything regarding the title of the paper. There should be a mention of ARNAX early in the paper if that remains the main focus of the paper (as suggested in the title). It is important to briefly establish what ARNAX is, what studies have been done on it to date and what conclusions have been drawn from those studies. This will be important to understand the rationale of the review paper.

5.     Figure 2: NK is not considered “adaptive immunity” in general although its innate activation displays cytotoxic activities. Adaptive cytotoxicity is mainly mediated by CD8 T cells. Also, indicate cross-presentation of CD8 T cells here if cross-presentation is a main point in this manuscript as implied in the title.

Lines 18 and 58: MAMPs stands for “microbe-associated molecular patterns”. Please correct this in the text.

Lines 20-21 and 342: Not all vaccines contain adjuvants. Please rewrite these sentences for correctness.

Line 117: Please add relevant references here that support the statement and elaborate more about vaccine adjuvants. Since the focus of the paper is on adjuvants, it would be better to provide specific examples of adjuvants to describe this section.

Lines 141-145 and 152-153: Avoid redundancy. Some concepts/sentences are repetitive. These sentences need to be rewritten to make them concise and succinct.

Line 179 and throughout the text: Abbreviations and their full forms should be written at the first appearance in the text.

Figure 2: Please make sure that everything is labeled appropriately.

Lines 287-292: Please include relevant references here.

Line 317-319: Is there an article to support this statement? Please include references here.

Author Response

Reviewer 3

Seya et al. review the role of adjuvants and the underlying mechanisms with a focus on cross-antigen presentation by dendritic cells. Additionally, the review discusses some of the issues facing the clinical development of novel adjuvants.

This is an interesting review highlighting the potential benefit of using a dendritic cell adjuvant, such as a TLR3 agonist, ARNAX, that is a non-inflammatory adjuvant candidate. The authors have previously carried out several pre-clinical studies on investigating the adjuvant effects of ARNAX in cancer, influenza and COVID-19 vaccines and have published several articles on ARNAX adjuvant. However, some of those key details are missing in this review and the importance/significance of ARNAX needs to be briefly defined at the onset of the paper (in the summary and introduction sections) to make a final recommendation. Additionally, the following points need to be addressed to improve the quality of the paper.

  1. Is the title reflecting the main concept of the review paper? The current title seems to reflect just one aspect of the entire paper. Suggest rewriting the title to better cover the contents in this review. 
  2. The title mentions “ARNAX vaccine”, which is misleading. ARNAX is a vaccine adjuvant, not a vaccine. Please rewrite for clarity.
  3. There should be a brief mention about ARNAX adjuvant in the summary section for coherence with the main review point in this manuscript.
  4. Line 53-54: Does not mention anything regarding the title of the paper. There should be a mention of ARNAX early in the paper if that remains the main focus of the paper (as suggested in the title). It is important to briefly establish what ARNAX is, what studies have been done on it to date and what conclusions have been drawn from those studies. This will be important to understand the rationale of the review paper.
  5.    Figure 2: NK is not considered “adaptive immunity” in general although its innate activation displays cytotoxic activities. Adaptive cytotoxicity is mainly mediated by CD8 T cells. Also, indicate cross-presentation of CD8 T cells here if cross-presentation is a main point in this manuscript as implied in the title.

Lines 18 and 58: MAMPs stands for “microbe-associated molecular patterns”. Please correct this in the text.

Lines 20-21 and 342: Not all vaccines contain adjuvants. Please rewrite these sentences for correctness.

Line 117: Please add relevant references here that support the statement and elaborate more about vaccine adjuvants. Since the focus of the paper is on adjuvants, it would be better to provide specific examples of adjuvants to describe this section.

Lines 141-145 and 152-153: Avoid redundancy. Some concepts/sentences are repetitive. These sentences need to be rewritten to make them concise and succinct.

Line 179 and throughout the text: Abbreviations and their full forms should be written at the first appearance in the text.

Figure 2: Please make sure that everything is labeled appropriately.

Lines 287-292: Please include relevant references here.

Line 317-319: Is there an article to support this statement? Please include references here.

Reply to the comments

We appreciate the comments raised by Reviewer 3, which are highly constructive. We have revised the manuscript according to these comments.

  1. The reviewer is right. The title was changed according to the comments.
  2. I agree to this comment. Is it OK with the title?
  3. Thank you for a good suggestion, which made the point clear.
  4. We found the point in this review clear by the addition of the overview on ARNAX in the introduction.
  5. The point on NK cells were reflected in the text (lines 139 and 274). The dendritic cell targeting by ARNAX up-regulates CD8+ T and NK cell function. However, how about other innate lymphocytes and NKT cells are unknown. These were reflected in the text and Figure 2.

Minor points

Lines 18 and 58: corrected.

Lines 20-21, Line 342: The reviewer is right. Corrections were made.

Line 117: The paragraphs of adjuvant-mediated modulation of Ab production and innate lymphocyte (including NK cell) activation were extensively revised with references. Thank you for this critical comment.

Line 141-145, Line 152-153: We have removed the redundancy and revised the paragraph. 

Line 179: We checked abbreviations and spelled them out.

Figure 2: Figure 2 were re-labeled. The figure may not be sufficient for the labeling. If this is the case, we could replace this figure with an alternative one. 

Lines 287-292: Appropriate references were compensated.

Line 317-319: We rewrote this sentence. The statement is supported by Alum’s inflammatory tendency not by its direct effect. We corrected this issue in the text.

Round 2

Reviewer 3 Report

Overall, the main concerns in the first round of revision have been addressed, that an overview of ARNAX adjuvant should be added in the summary and introduction sections to make the rationale of this review clear, and the authors have revised the text to that end. Additionally, the authors have made extensive revisions to address the other major and minor points raised in the first round.

There are still a few minor concerns in the text.

1. The new title seems to better cover the contents in this review. However, it might be better to reword the title as ‘Toward establishing an ideal adjuvant for non-inflammatory immune enhancement’.

2. Figure 2: Since CD8 T cells were included in the revised figure, please add a sentence about CD8 T cells in the description too. Also, in the first part of the figure, ‘other cells in the body’ seems to be slightly vague. Consider providing examples of such cells in the description. The black arrows indicate the inflammatory pathway induced by viruses, but viruses have not been labeled in the figure to make that connection clear. 

3. Please ensure that appropriate words have been used according to the context so that the meaning of the sentence does not change. One example is: in Line 26, the word ‘undissolved’ needs to be replaced with ‘unresolved’. The text could benefit from some editing to improve grammar and phrasing, but overall is readable.

Author Response

There are still a few minor concerns in the text.

  1. The new title seems to better cover the contents in this review. However, it might be better to reword the title as ‘Toward establishing an ideal adjuvant for non-inflammatory immune enhancement’.
  2. Figure 2: Since CD8 T cells were included in the revised figure, please add a sentence about CD8 T cells in the description too. Also, in the first part of the figure, ‘other cells in the body’ seems to be slightly vague. Consider providing examples of such cells in the description. The black arrows indicate the inflammatory pathway induced by viruses, but viruses have not been labeled in the figure to make that connection clear. 
  3. Please ensure that appropriate words have been used according to the context so that the meaning of the sentence does not change. One example is: in Line 26, the word ‘undissolved’ needs to be replaced with ‘unresolved’. The text could benefit from some editing to improve grammar and phrasing, but overall is readable.

Reply to the comments

   I appreciate the comments raised by the Reviewer. Since the comments 1 ad 2 were reasonable, I reflected the comments in the relevant portions in the text. I found additional editions required to improve the text. The text has been further extensively revised according to the comments 3.

   We are grateful to this reviewer for this opportunity.

Best wishes,

   Tsukasa